# Assessment of the Hematopoietic Differentiation Potential of Human Pluripotent Stem Cells in 2D and 3D Culture Systems

**DOI:** 10.3390/cells10112858

**Published:** 2021-10-23

**Authors:** German Atzin Mora-Roldan, Dalia Ramirez-Ramirez, Rosana Pelayo, Karlen Gazarian

**Affiliations:** 1Departamento de Medicina Genómica y Toxicología Ambiental, Instituto de Investigaciones Biomédicas, Universidad Nacional Autónoma de México, Cuidad de México 04510, Mexico; germanmora@iibiomedicas.unam.mx; 2Instituto de Oftalmología Fundación, Conde de Valenciana IAP, Cuidad de Mexico 04510, Mexico; 3Centro de Investigación Biomédica de Oriente, Instituto Mexicano del Seguro Social, Km 4.5 Carretera Atlixco-Metepec, Puebla 74360, Mexico; dalramz0224@gmail.com (D.R.-R.); rosana.pelayo.c@gmail.com (R.P.)

**Keywords:** hPSC, primitive hematopoietic differentiation, Wnt signaling, embryoid body, 3D culture

## Abstract

Background. In vitro methods for hematopoietic differentiation of human pluripotent stem cells (hPSC) are a matter of priority for the in-depth research into the mechanisms of early embryogenesis. So-far, published results regarding the generation of hematopoietic cells come from studies using either 2D or 3D culture formats, hence, it is difficult to discern their particular contribution to the development of the concept of a unique in vitro model in close resemblance to in vivo hematopoiesis. Aim of the study. To assess using the same culture conditions and the same time course, the potential of each of these two formats to support differentiation of human pluripotent stem cells to primitive hematopoiesis without exogenous activation of Wnt signaling. Methods. We used in parallel 2D and 3D formats, the same culture environment and assay methods (flow cytometry, IF, qPCR) to investigate stages of commitment and specification of mesodermal, and hemogenic endothelial cells to CD34 hematopoietic cells and evaluated their clonogenic capacity in a CFU system. Results. We show an adequate formation of mesoderm, an efficient commitment to hemogenic endothelium, a higher number of CD34 hematopoietic cells, and colony-forming capacity potential only in the 3D format-supported differentiation. Conclusions. This study shows that the 3D but not the 2D format ensures the induction and realization by endogenous mechanisms of human pluripotent stem cells’ intrinsic differentiation program to primitive hematopoietic cells. We propose that the 3D format provides an adequate level of upregulation of the endogenous Wnt/β-catenin signaling.

## 1. Introduction

In vitro generation of human hematopoietic stem cells from embryonic and induced pluripotent stem cells (hPSC) serves multiple beneficial purposes: mechanistic studies of hematopoiesis, development of cell therapy for hematological diseases, induced transplant tolerance, disease modeling, and drug screening, among others [1] [2]. The current protocols for the generation of hematopoietic cells are based on the recapitulation of developmental signals similar to those in the embryo. In vitro, hematopoietic progenitors originate from mesoderm through hPSC differentiation, as a part of the process that starts upon the exposure of hPSCs to the appropriate dose of bone morphogenic factor 4 (BMP4) [3]. Then the mesoderm begins to form the hemogenic endothelium resulting in hematopoietic commitment (Figure 1) [4,5]. Much effort has been applied to defining the appropriate differentiation conditions required to obtain hematopoietic stem cells in vitro; for example signal gradients [6], cytokines in suitable combinations with the appropriate timing and specific microenvironments, such as feeder cells or artificial niches and with different culture systems, such as embryoid bodies (EBs) [7]. The EB has been employed widely in studies of hematopoiesis to parallel the course of embryonic development and to investigate the primitive (and definitive) hematopoiesis represented by the hemangioblast [8,9,10] that produces erythroid cells and monocytes [4,5,11,12,13]. Alongside, a 2D monolayer has been considered to be adequate for the induction of primitive hematopoiesis [14,15,16,17,18]. In both EB and 2D Monolayer systems, exogenous Wnt/catenin pathway stimulation was used to enhance the definitive developmental compartmentalization/wave of hematopoiesis. Together, these studies contributed to the current concept of in vitro hematopoietic differentiation and raised questions regarding the reasons for using either EBs or monolayers to produce hematopoietic cells (See Table 1). Recent studies demonstrated the advantages of 3D organoid protocols of hPSC differentiation and generation of clinical-grade definitive human progenitor cells and stem cells of hematopoietic lineages, potentially engraftable lymphoid progeny for transplantation studies and, erythroid progeny capable of producing adult hemoglobin [19]. 3D culture technology has also been improved by the employment of biomaterials to create 3D tumor models for drug screening [20,21]. Additionally, when applied in ectodermal-derived lineages, the 3D format improved neuronal differentiation, and the formation of a neural network, recapitulating brain tissue-like environments suitable in a disease model-specific extracellular aggregation [22]. In the past, the comparative potentials of 2D and 3D culture platforms for supporting in vitro hematopoietic differentiation of hPSCs have been explored, to improve their pluripotency and escalate cell productivity [23,24].

In the same context, comparison between 2D and 3D conditions has been performed in studies of other differentiation models, such as osteogenic differentiation [25] and, widely in cardiac differentiation. Studies with hPSC-derived cardiomyocytes found that the differentiation potential was affected by the choice of culture platforms, either it is 2D or 3D [26,27,28,29,30]. This study aimed to reveal differences between the 2D and 3D conditions observable during the generation of primitive hematopoietic cells from hPSCs in the same media and cytokines. Previously, some differences between 2D and 3D settings in lineage specification processes, as well as its (3D) advantages, have been noted in independent studies [14,31,32,33]. However, in independent experiments, due to the extreme plasticity of the early embryogenesis, the microenvironment differences can influence distinctly the developmental processes hence only by the use of the same experimental setting, same culture condition, parallel assays, and principles of interpretation can we highlight reliably, the intrinsic differences between the two culture methods observed at each of the steps of the in vitro induction and differentiation. To address this issue, we have undertaken comparative experiments of 2D and 3D systems regarding the capacity of each of them to support the primitive hematopoietic differentiation using a commercial embryonic stem cell line and an induced pluripotent stem cell line generated in our lab (Appendix A).

## 2. Materials and Methods

### 2.1. Embryonic Stem Cell and iPS Cell Lines

The ES cell line ESI-017 was purchased from ESI-BIO (Alameda, CA, USA). Induced Pluripotent Stem Cells were generated as described previously [34]. Briefly, human dermal fibroblasts (ATCC catalog number PCS-201-010) were transduced with lentiviral vectors containing the genes encoding reprogramming factors Oct 4, Sox2, Nanog, Klf4 and, C-Myc resulting in the formation of colonies on day 15 to 20 days post-infection. Colonies were then selected according to morphology and compaction and expanded to passage 30 and maintained until ready for differentiation. The use of human cells in this study has been approved by the Ethics Committee of the Biomedical Research Institute of the National Autonomous University of Mexico (UNAM).

### 2.2. Cell Culture under 2D/Monolayer and 3D/Embryoid Body Conditions

Human ES and iPS cell colonies were cultured as 2D/monolayer and 3D/EB in Essential 8 medium, E8 (Gibco; Thermo Fisher Scientific, Inc., Waltham, MA, USA, then detached using 0.1% Type IV Collagenase (Sigma-Aldrich, St. Louis, MI, USA) followed by mechanical disaggregation. The 2D/monolayer cells were seeded at a density of 1 × 10^4^ onto Matrigel () (MG)-coated plates (Corning, NY, USA) whereas for 3D culture untreated, ultralow attachment plates (Nunc, Rochester, New York, USA) were used. Cell clumps were maintained in E8 medium until spheres were grown to no more than 300 μM in diameter. Obtained spheres were passaged after collagenase treatment and mechanical disaggregation. Cell clumps of around 100 to 200 µm were seeded and cultured in the same conditions until initiation of differentiation (day 0).

### 2.3. Mesodermal, Hematoendothelial and Hematopoietic Differentiation

To observe the effects of the two (2D and 3D) culture systems on differentiation, neither Wnt pathway activators nor inhibitors were added to the media as conventionally done (See Table 1). The differentiation of the hPSCs took place in 12-well plates (Nunc) treated with MG for the 2D and in ultra-low attachment non-treated plates for 3D. Mesodermal differentiation proceeded in Stempro 34 (Sigma-Aldrich) medium supplemented with 10× ITS (Gibco), 40 ng/mL BMP4, 10ng/mL FGF2 (Preprotech), 50 μg/mL Ascorbic Acid (Sigma-Aldrich), 100× Glutamax (Gibco) and Gentamicin (Gibco) from day 0 to day 4. On day 4, the medium was changed to Stempro 34 with 10× ITS, 100× Glutamax, Gentamicin, 10 ng/mL FGF, 10 ng/mL VEGF; 10 ng/mL of IL6 and 2 U/mL EPO (Preprotech) for three more days (days 4 to 6). On day 6 the medium was replaced with hematopoietic differentiation Stempro 34 medium containing the above supplements plus 50 ng/mL SCF, 5 ng/mL IL7, 5 ng/mL Flt3 and 10 ng/mL IL3 (Preprotech Rocky Hill, USA). CHIR99021 (3 μM) and IWR-1(1 µM) (Sigma), were added only when indicated.

### 2.4. Hematopoietic Colony Assay

Analysis of hematopoietic colony formation potential of the hematopoietic stem cells was performed by plating 2 × 100 cells in Methocult (Stem Cell Technologies Inc. Vancouver, Canadá) supplemented with SCF (100 ng/mL), EPO (2 U/mL), IL-6 (5 ng/mL), IL-3 (40 ng/mL), TPO (40 ng/mL) (Preprotech). Colonies were quantified after 30 days. Cells were separated from the Methocult by centrifugation and fixed for Wright staining.

### 2.5. AFT024 Co-Culture for Further Hematopoietic Differentiation

To generate further hematopoietic lineages, we employed co-culture with the AFT024 cell line, which is derived from mouse fetal liver stroma, this cell line expresses DL1, an important factor during the acquisition of advanced hematopoietic fates. Cells were purchased from ATCC (SCRC-1007) and transduced with a lentiviral GFP reporter (pLenti CMV GFP Puro, Addgene) [34]. Passage 4 cells were expanded with Optimem (Gibco) and 5% FBS (Gibco) until confluency and then seeded at 10,000 cells per well in 96 well plates. On days 6 to 8, 2D/monolayer and 3D/EB cells were plated on the AFT024 monolayers with Optimem medium supplemented with 1%FBS, 100X Glutamax, and IL7, IL3, SCF with a change of monolayer every 5 days during 30 days [35].

### 2.6. Flow Cytometry Analysis

Flow cytometry analysis was done in a FACSCanto cytometer (Becton Dickinson, NJs, USA). Fluorophore-conjugated antibodies used for flow cytometry immunostaining are listed in Appendix A. Immunostaining was performed as recommended in the technical datasheet for each antibody. Analysis of flow cytometry acquisitions was performed using FACSdiva Software (Becton Dickinson, BD Biosciences, CA, USA) and FlowJo v10.0.7 (FlowJo, LLC, BD Biosciences, CA, USA). Ten thousand events were recorded for each sample. Analysis of Flow Cytometry Data and the fold increase in mean fluorescence was performed with FlowJo v10.0.7.

### 2.7. Immunofluorescence

Cells were seeded on MG-coated chamber slides (Nunc-Labtek, Thermo Scientific, USA). Cells were fixed with 4% PFA for 15 min at room temperature (RT) and then permeabilized with 0.2% Triton in PBS for 10 min. Cultures were blocked using 1% BSA plus 0.2% Triton in PBS for 30 min at RT and then incubated with primary antibody in blocking solution for 1 h at RT or overnight at 4 °C. After washing, the secondary antibodies were added, washed and cells were put in blocking solution for 1 h at RT in the dark. After washing, nuclei were stained with Hoechst 33342 (Invitrogen, CA, USA). Epifluorescence analysis was performed using an Olympus IX71 phase-contrast fluorescence microscope and images were analyzed with QCapture Suite software (QImaging, Surry, BC, Canada). Microscopy analysis was performed using a Nikon confocal microscope and images were analyzed with the Image J (ImageJ, U. S. National Institutes of Health, Bethesda, Maryland, USA) software, Z Stacks of the 3D cultures were obtained using the Bioformats plugin. The list of antibodies used and the catalog number is available in Appendix A. 

### 2.8. Gene Expression Analysis by RT-qPCR

Total RNA was isolated using TRIzol reagent according to the manufacturer’s instructions (Invitrogen, CA, USA) followed by DNase I (Invitrogen, Waltham, MA, USA) treatment and an RNA clean-up using the RNeasy mini kit (Qiagen, Hilden, Germany). Reverse-transcription reactions were performed using a One-step RT-PCR kit (Qiagen). Quantitative RT-PCR (RT-qPCR) reactions were performed using KAPA SYBR FAST (Roche) master mix on a Rotor-Gene 6000 thermal cycler (Qiagen). Data analysis was performed using Rotor-Gene Q series Software (Qiagen) and relative expression was calculated using Pfaffl’s efficiency calibrated method [36]. The primers for the PCR experiments are listed in Appendix A. 

### 2.9. Statistical Analysis

Values are given as mean with SE, and statistical significance was performed from at least three independent biological replicates. Student’s *t*-test or analysis of variance (ANOVA) were performed to analyze two or more groups of means, respectively, using GraphPad Prism version 6.04 for Windows, (GraphPad Software, La Jolla, CA, USA). Tukey’s post hoc test was performed to determine the statistical significance of the ANOVA.

## 3. Results

### 3.1. 3D/EB and 2D/Monolayer Culture Conditions Differ in Their Capability to Produce Hematopoietic Cells

Our main objective was to compare the capacity of these formats to support the cell commitment events during differentiation of hPSC to primitive hematopoiesis in the lack of exogenous stimulation of Wnt/β-catenin shown previously to direct differentiation to definitive hematopoietic lineages [37,38,39,40,41,42]. To this purpose, we performed simultaneous differentiation in these formats, with the same culture environment and assay methods, without the addition of any Wnt/β-catenin exogenous stimulator like CHIR99021. Figure 2A depicts the cells’ morphological changes in 2D and 3D formats on days 0, 4, 6, and 8 of hPSCs differentiation driven exclusively by endogenous factors. Before differentiation, we tested the capacity of the protocol we used to ensure similar expanding and pluripotency properties of the passage 30 hESC and hiPSC, by seeding them in the same media as a monolayer at a density of 1 × 10^4^(2D) or as spheres (3D) (see the section of Materials and Methods) and culturing for 5 days. The Immunofluorescence analysis with antibodies to Tra160, Oct4 and NANOG showed no significant differences between the two cultures, evidencing that hPSCs used in the study were similar from the start of the differentiation, hence the differences that could arise will be from the specific effects of the culture formats (Figure 2B) (Appendix A).

As the pluripotency of hPSCs was similar in both 3D and 2D colonies, we set on to find possible differences in the hematopoietic commitment of the cells under the two culture conditions. As was reported (see references in Table 1), early hematopoietic commitment can be identified starting from Day 6 through detection and quantification of a cell cohort with the CD34+/(Kinase domain receptor), KDR+ phenotype. Although CD43 would be adequate for the identification of hematopoietic populations in these experiments, yet we found the CD34+/KDR+ co-expression preferable to identify cells in transition from mesoderm to hematopoietic and endothelial phenotypes. With this intention in mind, we measured the number of CD34+/KDR+ cells by flow cytometry and compared the numbers of CD34-positive cells under the two culture formats on days 4, 6, and 8 (Figure 3A) (Appendix A). A higher number of CD34+/KDR+ cells in the 3D/EB condition was found. Additionally, we measured the colony-forming capacity of cells from both systems grown on methylcellulose for 30 days (Day 6 + 24). For this purpose, day 6 cells were divided in half, one of which was used for FACS analysis and the other half used in the methylcellulose colony formation experiments. (Figure 3B). Two hundred cells (*n* = 3) from each of the formats were used and obtained on average seven colonies (four of them are shown in Figure 3). Although CFU’s were already detectable at 10–15 days, we continued to culture up to 30 days (6d + 24d), cells were isolated and then wright-stained and identified as CFU-GM colonies. Clear differences between the two culture formats were revealed: round and refractive groups of cells were found in the 3D/EB while monolayer/2D cultures did not show colony-formation (Figure 3B). We did not separate and identify more CFU types, considering that only one type already demonstrated a sharp difference between the two formats. As there were no colonies at the 2D/monolayer condition (*n* = 3), the quantitative determination of CFU capacities was not performed.

The cytometry analysis results described in Figure 3B showed that only the suspension (3D/EB) culture format produced a population of CD34+KDR+ cells. When further analyzing this double-positive subset, we found that this cell population expresses almost in its entirety CD31 (97.5%). This suggested the emergence of a hemogenic endothelial population (CD34+KDR+CD31+). Simultaneously, we found evidence (lower panel) of a population that is positive for CD34, and positive for CD31, but negative for KDR, a possible indication of non-hematopoietic endothelial cells, showing that hematopoietic and non-hematopoietic cells can arise simultaneously during this period of differentiation (44–47). This finding correlates with the formation of colonies in the 3D and the lack of colonies in the monolayer format. Immunofluorescence showed on day 6 the co-expression of CD34, KDR, and CD43 in the 3D but not in the 2D format (Figure 3C). These results indicated that the 2D/monolayer system was significantly inferior in supporting the hematopoietic differentiation, basically, and specifically, in the development of clonogenic CD34 hematopoietic cells. Additionally, to know whether the observed distinctions were due to a slower differentiation of 2D/Monolayer cells, we compared the time course of differentiation of the cells in the two formats at later time points (Day 8 and Day 10) (Figure 3A). The results showed a consistent time course pattern of CD34+ cell generation on days 8 and 10 (Appendix A), indicating that there was not a higher number of positive cells at any time point during the differentiation of the 2D/Monolayer. Once the above differences between the two systems were observed, we sought to find out the earliest time point at which these differences became apparent. To this end, we performed qRT-PCR analysis of the expression of day 4 cells of key mesodermal commitment markers: Brachyury (T), BMP, KDR, Plate derived growth factor receptor (PDGFR), and Sox17, a marker associated with endodermal lineage but also known as a differentiation-related factor during the acquisition of endothelial phenotype required for the early hematopoietic fate establishment [43,44,45,46]. We found that the mesoderm was phenotypically well established in 3D/EB by day 4; in contrast, the 2D format cells showed a reduced or no commitment (Figure 3D) (see the primers in Appendix A). In particular, the expression of Brachyury (T) was two-fold higher in 3D/EB system than in the 2D system (*p* ≤ 0.001). We analyzed the expression of the *KDR* gene, an important mesodermal marker that, unlike the *T* gene, remains active in cells undergoing hematopoietic commitment [45]. In our experiments, the 3D/EB mesoderm cells showed a high expression of KDR, whereas the 2D/monolayer mesoderm cells were almost negative for this marker, suggesting that none or very few of these cells were competent for hematopoiesis. PDGFR has been used to identify lateral plate mesoderm and to distinguish it from paraxial mesoderm [47,48]. We found that the level of expression of the *PDGFR* gene by 3D/EB cells exceeded more than twofold its expression by the 2D/monolayer cells (*p* ≤ 0.001), suggesting that the 2D system had a weaker capacity of producing lateral mesoderm. The expression patterns of all these markers suggested that the 2D/monolayer system was deficient for the appropriate induction in the mesodermal cells (positive for T and KDR) of commitment to the hematopoietic lineage (KDR), displaying, at the same time, a strong selectivity for lateral mesoderm (PDGFR) commitment to early (primitive) hematopoiesis (So×17). Figure 3E compares T and KDR immunofluorescence staining on day 4. Additionally, in line with the above results (Figure 3D), we found that the mesodermal marker Brachyury (T) was expressed by cells of both formats, but to a different extent. Commitment was distinctively achieved in the two systems. However, as there was little or no KDR staining in 2D cells, it can be posited that 2D/Monolayer condition could support a partial mesodermal commitment but is still insufficient to fully achieve hematopoietic specification. In common, these results suggested a weaker capacity to induce lateral mesoderm of the 2D system, thus resulting in a diminished hematopoietic commitment.

### 3.2. Endogenous Wnt/Catenin Signaling Is Active in the 3D/EB System but Not in 2D/Monolayer System

Since the 3D/EB culture system displays an enhanced mesodermal and hematopoietic commitment, we sought the possible causes of this phenomenon. Our group has previously reported that Wnt/β-catenin is an essential regulator of the biphasic human somatic cell reprogramming [34]. In particular, Wnt activates mesendodermal genes during colony maturation towards the pluripotent state. Furthermore, in many studies [17,41,49] exogenous Wnt/catenin was used to stimulate the production of hematopoietic populations from hPSCs. The comparative cytometry assays revealed that β-catenin was abundantly expressed by the 3D/EB and was almost non-expressed by the 2D/monolayer cells (Figure 4A) (Appendix A). In concordance with this higher presence of β-catenin in 3D cells, Figure 4B shows the higher expression in these cells of the *Axin2* gene, one of the Wnt transcription factors. Figure 4C shows immunostaining of the markers T and β-Catenin, it is observed that in the 3D condition both markers are present, however, in the monolayer condition, only T was present. These findings suggested that the high level of hematopoietic commitment in mesodermal cells and a more efficient differentiation to CD34+ cells with clonogenic potential in the 3D/EB format might be due to the higher endogenous levels of Wnt/catenin signaling. In line with this suggestion, the 2D/monolayer system’s inability to sufficiently activate endogenous Wnt/catenin signaling might be a cause of its relative inefficiency to induce the commitment to the hematopoietic fate. To complement these data, we examined the two culture platforms for the ability to generate cells with advanced hematopoietic commitment; to this end, we sought to obtain lymphoid lineages by co-culturing the cells with AFT024 stromal cells as a feeder, providing in vivo-like microenvironment for advanced hematopoietic differentiation. Appendix A compare the two culture formats regarding the presence of CD56, which is most stringently associated with, but certainly not limited to, natural killer cells. As expected, the 3D format produced a higher number CD56 -positive cells than did the monolayer format on day 30 of culture.

Once it was shown that the 3D/EB system provides higher endogenous Wnt activity, we tested the effects of adding CHIR99021, a Wnt pathway activator, or IWR-1, a Wnt pathway inhibitor (Appendix A). The rationale for this was to look at the effects on hematopoietic differentiation of additional (to the endogenous) level of Wnt/β-catenin induced by CHIR99022 in the 3D/EB cells (Figure 4D). It is known [50] that endothelial cells (ECs) are generated from mesoderm during hPSC differentiation, and are comprised of two sub-populations: non-hemogenic (vascular) and hemogenic endothelium; it is also known that the CD34+/CD31+ population is a hemogenic endothelium with primitive potential [51]. As was described above, we split the EB+CHIR derived cells into two portions, one of which was cultured in CFC and the other one subjected to FACS. We found that the 3D cells acquired an advanced clonogenic capacity evidenced by the generation of erythroid colonies (CFU-E) (Figure 4D) (Appendix A). In Figure 4E, we show that activation of Wnt by CHIR99021 depleted the CD31+ subpopulation that was present in the 3D/EB condition without Wnt activation. Additionally, activation of the Wnt pathway depleted the CD31+/KDR+ fraction that forms a part of the endothelial- primitive hematopoietic lineage cells (Figure 4E). Our results on the depletion of these populations correlate with the published (Table 1) data, indicating that exogenous Wnt activation is a means to launch both primitive and definitive hematopoietic programs. 

Previously, Keller’s and Slukvin’s groups (see Table 1) demonstrated that the mesoderm gives rise to a hematopoietic CD34+/CD43− population detectable on day 6 of hPSC differentiation. This population expresses the markers of hemogenic endothelium and possesses multi-lineage (definitive) hematopoietic potential. On the other hand, the CD34+/CD43+ population may represent primitive hematopoietic cells. We found that when 3D/EB cells were treated with CHIR, the CD43+ fraction was absent in the day 6 cultures. We consider that our EB+CHIR Wnt activation condition depleted the primitive CD34+CD43+ in favor of the CD34+CD43− phenotype, in line with Keller’s finding. Conversely, this population remained stable in untreated 3D/EB cultures. Altogether, these results further support the notion of the 3D/EB condition’s higher capacity to mimic the early embryonic events (primitive wave of hematopoiesis) during in vitro differentiation (Figure 4E). In contrast, the expression levels of the markers CD34, CD31, KDR, and CD43 were low in the EB+IWR cells, providing additional confirmation that the 2D/monolayer is by itself insufficient to generate hematopoietic cells (Figure 4D,E).

## 4. Discussion

In previous studies (e.g., the references in Table 1) with the 3D/EB culture format, it was shown that in the presence of growth factors, specific cytokines, and other ingredients, hPSCs exposed to appropriate doses of BMP4 [38] were committed through activation of T (Brachyury) and KDR genes to the mesoderm, then through hemogenic endothelium, to transient primitive hematopoietic bipotential hemangioblast cells capable of differentiating into hematopoietic and endothelial cells. On the other hand, in independent experiments using the 2D format, hPSCs required exogenous stimulation of Wnt signaling for the generation of definitive hematopoietic populations. The specific contribution of each of these two formats is difficult to evaluate in independent experiments, because the distinctions in the microenvironment factors, different assay conditions and differences in interpretation of results could obscure their effects on in vitro hematopoietic processes. This study compares the potentials of these two formats to support the hPSCs hematopoietic differentiation in the same experimental settings without exogenous stimulation of Wnt signaling. Our results describe in detail how each format supports hematopoietic differentiation, starting from the mesoderm formation and commitment to the posterior stages up to the clonogenicity stage. To our understanding, this is the first such study whose results contribute to the estimation of the potential of the endogenous machinery to realize the differentiation program in two experimental settings and use these results in the conceptualization of in vitro hematopoiesis. Our results support the view that the 3D/EB system efficiently mimics the in vivo embryonic events. In addition, our results provide a tentative explanation for the higher capacity of the 3D/EB system, by showing that endogenous Wnt/catenin signaling is more active in this format on day 4 [39,41,49]. The novelty in our findings is the demonstration of the importance of adequate activation of endogenous Wnt signaling by BMP in the 3D format, which is required for robust mesoderm formation through the activation of the genes encoding T and KDR, which determines successful commitment to hemogenic endothelial precursors of the clonogenic CD34 hematopoietic cells. In contrast, the 2D/monolayer system displays a considerably lower level of endogenous Wnt/catenin (Figure 4) and therefore, a lower number of hematopoietic CD34 cells which lack clonogenic capacity. According to the results of this comparison, it can be hypothesized that the 2D/monolayer system deficiency results from its incapacity to respond adequately to BMP4 for complete mesoderm formation. To compensate for this weak potential, exogenous stimulation of Wnt signaling is practiced, but this directs differentiation to the definitive hematopoietic fate. Significantly, these deficiencies of the 2D format reflect the impossibility of such a format to model the unique, in vivo, 3D-like embryonic development. Meanwhile, the 2D format is capable to serve as a biotechnological platform for the production of hematopoietic cells useful for therapeutic purposes, in which cell-autonomous factors can operate only, leaving in an inactive state the integrative regulators of the developmental mechanisms. 

Finally, this study shows the clear advantages of 3D culture as a basis for the advances in the understanding of the mechanistic aspects of in vivo hematopoietic events. Prospectively, this capacity of the 3D/EB format to model adequately the 3D niche condition for primitive hematopoiesis can also help to explain anomalies caused by mutations in the KDR and GATA2 genes, thus proposing an adequate in vitro system for drug testing [52,53].

## Figures and Tables

**Figure 1 cells-10-02858-f001:**
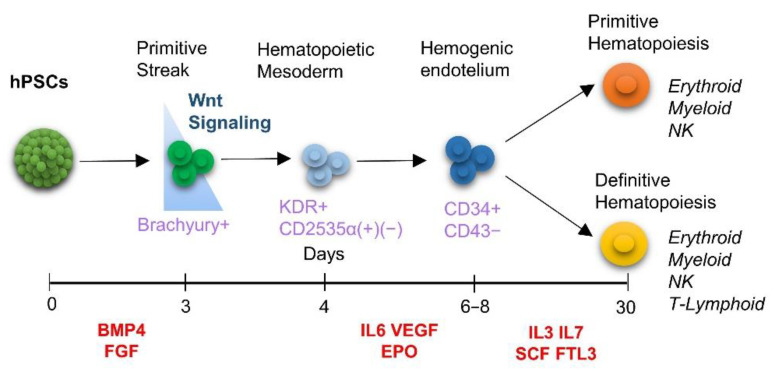
Schematic representation of current knowledge of in vitro differentiation of human pluripotent stem cells to hematopoietic lineage. Hematopoietic differentiation of hPSCs is depicted based on the protocols in the literature. Hematopoietic differentiation occurs in a consecutive stage-specific manner, starting with the BMP4-induced mesodermal commitment (days 0–4) followed by conventionally employed exogenous activation of Wnt/β-catenin signaling at this initial stage. The Lateral plate (Day 3–4) mesoderm generates endothelial progenitors (4–6), hemogenic endothelium with the potential to hematopoietic lineages (Day 6–8). At this latter stage, the population splits into two developmental programs/waves: primitive hematopoiesis, characterized by the formation of monocytic cell types, and definitive hematopoiesis with lymphoid potential. Abbreviations: FGF: Fibroblast Grow Factor, IL3: Interleukin 3 IL6: Interleukin 6, IL7: Interleukin 7, VEGF: Vascular Endothelial Grow Factor, EPO: Erythropoietin, SCF: Stem Cell Factor, FLT3: receptor-type tyrosine-protein kinase.

**Figure 2 cells-10-02858-f002:**
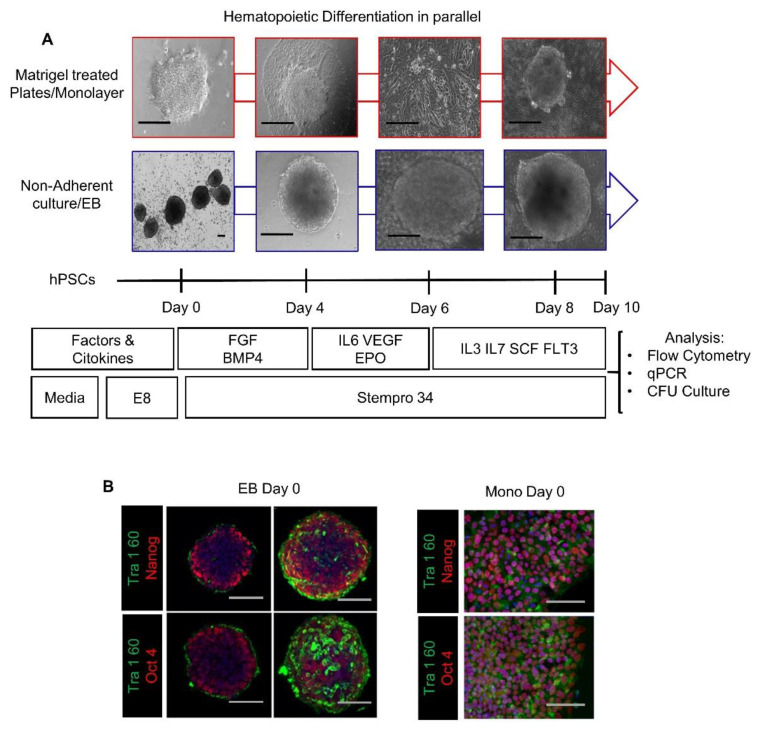
Comparison of the pluripotency of hPSCs in 2D/monolayer and 3D/EB culture systems. (**A**) Experimental procedure of in vitro differentiation in parallel settings showing representative phase-contrast images of cells at Day 0, Day 4, Day 6 and Day 8. (**B**) Immunofluorescence analysis of 3D/EBs clones and their cross-sections, and 2D/Monolayer cells, stained by antibodies to Tra-160, Oct4 and NANOG. Scale Bars: EB: 100 μM, Monolayer: 100 μM.

**Figure 3 cells-10-02858-f003:**
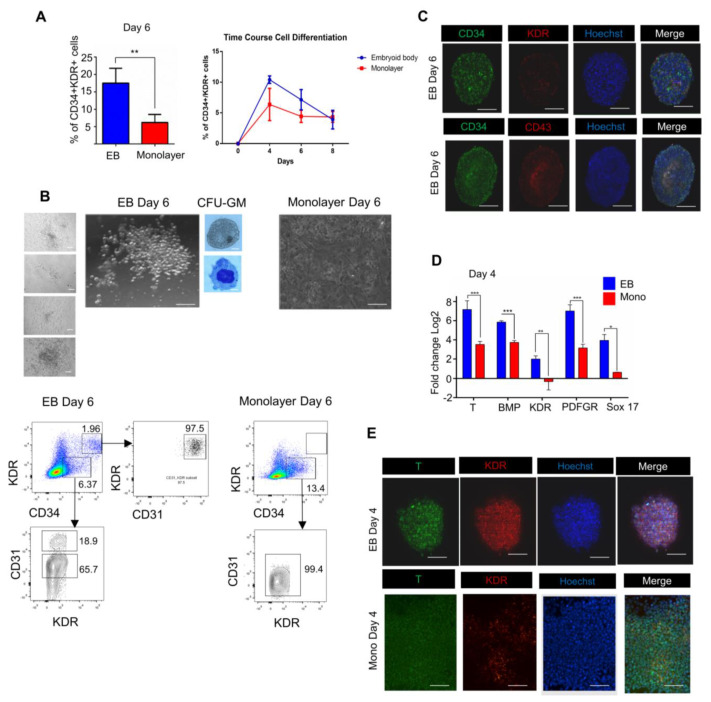
3D/EB suspension and 2D/monolayer formats differ significantly in their capability of producing hematopoietic cells from hPSCs. (**A**) Comparison of the percentages of the cells with CD34+/KDR phenotype generated by the 3D and 2D systems on day 6 (Mean ± SEM) *n* = 4, ** *p* ≤ 0.01. Time course of CD34/KDR cells in 3D and 2D formats (**B**) Phase contrast image of hematopoietic colonies in methylcellulose shown for 3D/EB on day 30 after Wright-Staining (see methods), showing images of a typical macrophage and a monocyte. No such cells were present in the 2D system. Flow cytometry analysis of the day 6 3D/EB and 2D/monolayer cells for co-expression of CD34 and KDR (without sorting of CD34 cells) rectangles indicate the populations of interest with antibody stain. Abbreviations: CD34: Cluster of differentiation 34, CFU-GM: Colony-forming unit Granulocyte-Macrophage. (**C**) Immunofluorescence of hematopoietic markers CD34, KDR and CD43 on Day 6 of the 3D/EB cells; images were processed with Image J to obtain sections (z-stack) of EB using the bio formats plug-in (see methods). Abbreviations: T: Brachyury. (**D**) Quantitative RT-PCR determined expression levels of mesodermal genes on day 4; bars show the expression levels of the genes relative to the *GAPDH* gene; (Mean ± SEM) *n* = 3, * *p* ≤ 0.05, ** *p* ≤ 0.01, *** *p* ≤ 0.001. Results of three independent experiments with RNAs of iPS and ES cells are shown. (**E**) Immunofluorescence of mesodermal markers T and KDR on day 4 for 3D/EB and 2D/Monolayer at 20×, scale bar: 100 μm.

**Figure 4 cells-10-02858-f004:**
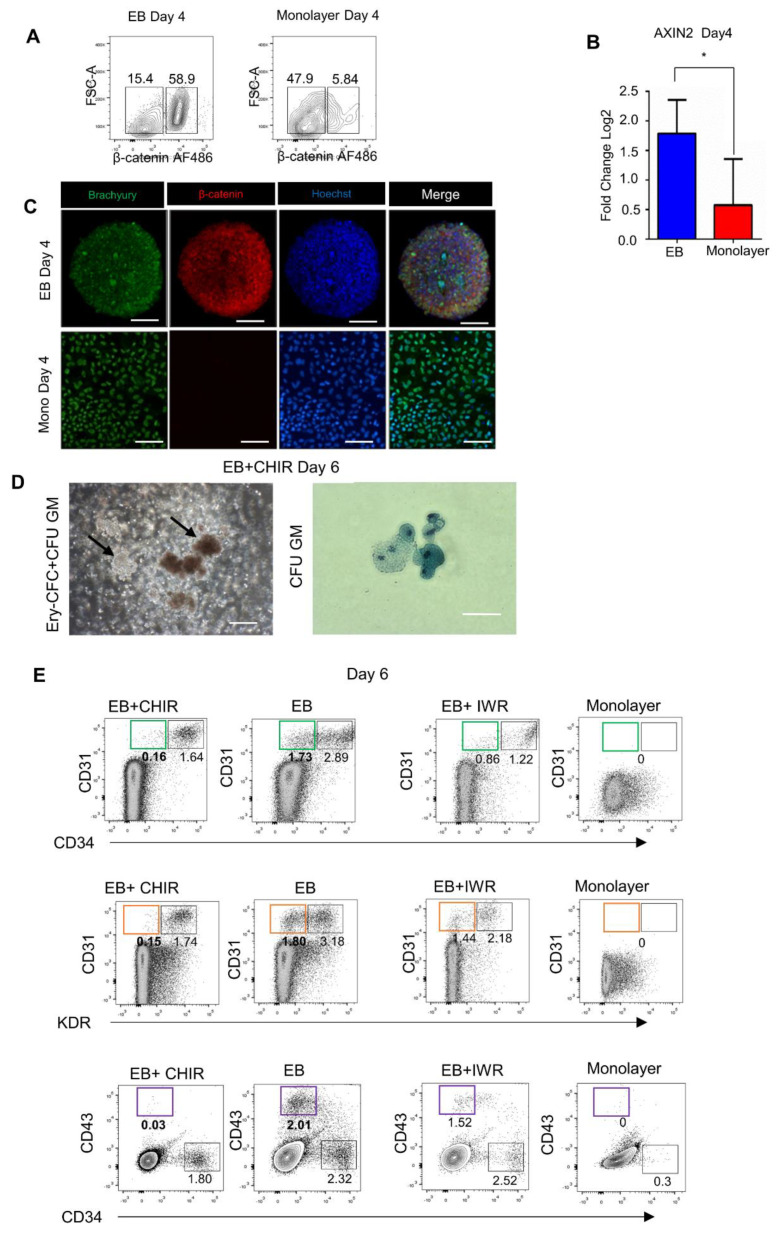
Endogenous activity of Wnt/β-catenin signaling in 3D/EB and 2D/Monolayer cells on day 4 of PSC differentiation. (**A**) Representative flow cytometry analysis of Day 4 cells, comparison of β-catenin-positive cells determined by FACS, and (**B**) Quantitative PCR analysis of Axin2, and (**C**) Immunofluorescence of Brachyury and β-catenin. Scale Bars: 100 μm, (Mean ± SEM) *n* = 3, * *p* ≤ 0.05. (**D**) Phase-contrast images of disaggregated 3D/EB+CHIR day 6 cells cultured in methylcellulose; arrows indicate colonies of EryCFC and CFU-GM wright-stained. (**E**) Representative flow cytometry analysis of Day 6 3D/EB cells after the use of Wnt activator/inhibitor showing the results of the treatment either with CHIR or with IWR, green rectangle shows the population of interest, the endothelial component in CD31/CD34 cells, orange rectangle shows the endothelial component in the CD31/KDR cells and purple rectangle distinguishes the primitive hematopoietic component in the CD43/CD34 cells. Source of the cells: CD34/KDR populations (**D**), CD31/CD34, KDR/CD31, and CD34/CD43 populations.

**Table 1 cells-10-02858-t001:** Selected publications on the independent use of 3D/Embryoid Body and 2D/monolayer systems in hematopoietic differentiation from hPSCs. The table displays the differences in the generation of hematopoietic lineages under these two conditions. Additionally, results show the importance of Wnt/β-catenin signaling activation during in vitro differentiation of hPSCs of the definitive hematopoietic fate.

Study	hPSCs	Culture System	Day(s) Hematopoietic Commitment	Hematopoietic Wave	Lineage Cells	Wnt Activation
Yanagimachi et al., 2013	hESCs, hiPCSs	2D/Monolayer	6	Primitive	Monocytes, Dendritic Cells	No
Sturgeon et al., 2014	hESCs, HiPSCs	3D/EB	6	Primitive definitive	Monocytes, Lymphoid Lineages	Yes
Ruiz et al., 2019	hiPCSs	2D/Monolayer	5–9	Primitive	Primitive Hematopoietic Cells	Yes
Niwa et al., 2011	hiPSCs	2D/Monolayer	6	Primitive	Monocytic Lineages	No
Kennedy et al., 2007	hESCs	3D/EB	5–6	Primitive	Early Hematopoietic Progenitors	No
Galat et al., 2017	hiPSCs	2D/Monolayer	5	Primitive	Monocytic lineages	Yes
Ditadi & Sturgeon, 2015	hESCs	3D/EB	8–15	Primitive, Definitive progenitors	Hemogenic Endothelium and early hematopoietic progenitors	Yes
Knorr et al., 2013	hIPSC, hESCs	3D/EB	11	Definitive	Nk, Lymhpoid	No

## Data Availability

Not applicable.

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
