# Peer review of "Assessment of the Hematopoietic Differentiation Potential of Human Pluripotent Stem Cells in 2D and 3D Culture Systems"

_cells, 2021, doi:10.3390/cells10112858_

Round 1
Reviewer 1 Report
This study compares the potential of two widely used in vitro culture systems, 3D/EB and 2D/monolayer for the induction of differentiation of human hPSCs towards the primitive hematopoietic lineage, which provide a comprehensive insight into the advantages of 3D/EB culture system. However, this study lacks a certain degree of innovation and further optimizes the 3D/EB system is necessary and meaningful. There are some problems in writing, data analysis and experiment design.
Major:
- Whether the 2D/monolayer and 3D/EB pictures are in the correct order in figure 2A,the picture of 3D/EB is more like growing in a plane.
- In Figure2, all annotations are unclear. there is an explanation of the D diagram in the figure2 comments, but there is no D diagram in the above figure.
- In Fig. S2, The flow analysis methods between the two groups are not consistent. Is the data obtained in this way accurate? In the flow chart below, there is the same problem.
- In line 270, “3D/EB cultures produced two fold higher number of the CD34+ hematopoietic cells than the 2D/monolayer system.” while in figure3B, the data was the opposite.
- in line 306, Why only T and KDR genes were selected for immunofluorescence staining.
- The high expression of Axin2 gene and β-catenin cannot directly draw this conclusion” the high level of hematopoietic commitment in mesodermal cells and differentiation to CD34+ cells with clonogenic potential in the 3D/EB system was due to the higher endogenous level of Wnt/catenin signaling.” There is no more data to support it.
Minor
- Abbreviations need to be defined when they first appear in the text (eg. Line 35 induced pluripotent stem cells, Line 66).
- In Materials and Methods part, cytokines concentration addition were ng/μl. Please make sure the unit are corrected.
- in line 132, the function of AFT024 should be explained and what kind of cells it belongs to?
- 1.in line 174-175, “pathway activation molecules” refer to the activator of Wnt?
- The experimental results should be rigorous. In line 189, the word “at glance” is not appropriate.
- In the text, it always appears as KDR, but in the data of figure3A, it is expressed as CD309. Although the two are the same gene, they need to be unified for the rigor of the article.
- When testing the relevant indicators, how to consider the selection of the number of days?
- In the supplementary picture, some pictures are not clear and need to be modified.
Reviewer 2 Report
In manuscript cells-1380870, Mora-Roldan et al. compare 2D/monolayer versus 3D/embryoid body-based hematopoietic differentiation of pluripotent stem cells without WNT pathway manipulation. They found their 3D/EB differentiation protocol to be more efficient in generating hematopoietic cells than 2D/ML.
In my opinion, this manuscript neither reveals interesting novel insights into the differentiation process nor describes methodological improvements. The manuscript is sloppy, many typos, inadequate / missing items or other errors disturb the flow of reading. I think the few conclusions that the authors draw from this study are speculative e.g. in lines 302-313, and are not of great value for the scientific community.
In the beginning, I want to point out that 2D cultures only initially grow in monolayers but later also in 3D, forming domes and vessel-like structures as can be seen also in Figure 2A (in which, by the way, captions of the upper and lower panels seem to be interchanged?!). Thus, also in the 2D cultures, cell-cell contacts and gradients of secreted morphogens (such as WNTs) may induce tissue patterning, but supposedly differently from EBs.
Major points:
1) The expression of CD34+/KDR+ indicates endothelial but not necessarily hematopoietic progenitors (lines 243-245 and 269-271). Agreed, these constitute a prerequisite for formation of hemogenic endothelium, but e.g. CD34+/CD43+ as in Fig. 3C would probably serve as a better-suited marker combination for hematopoietic readout.
2) MethoCult assays have to be quantified (usually done after 14 and not 30 days): how many colonies of which type were counted per how many seeded cells? How were the cells harvested/isolated before seeding? The monolayer condition microphotograph in Fig. 3B indicates MethoCult overgrowth by mesenchymal cells that should be excluded from seeding. The authors may also quantify the total number of live hematopoietic cells harvested from MethoCult to estimate average colony size. The authors should present representative pictures for all occurring CFU-types (CFU-G, -M, -E, -GM, -GEMM) and stains of a couple of cells not only one (Fig. 3B).
3) Several groups reported successful hematopoietic differentiation of hPSCs also in 2D culture without WNT manipulation. These studies demonstrated that 2D protocols can support development of clonogenic hematopoietic progenitors but the one chosen for this study did not. Thus, the observations in this paper do not allow general conclusions about superiority of embryoid bodies due to the single medium / time setting.
Minor points:
1) The percentage relationships (to the parent, grandparent, total?) in the flow cytometry gates are unclear. For example, what does the value of 0.6 in line 271 mean? The CD31+ gate contains certainly more than 0.6% of the parent CD34+/KDR+ population in Fig. 3B. The TRA1-60+ percentage of only ~50% would be low for undifferentiated hPSCs (Fig. 2B), but it appears significantly higher in the Fig. S1D histogram.
2) The conclusion in line 297 / Fig. 3D is not necessarily correct: the fact that KDR mRNA levels of the cell bulk are lower than that of GAPDH (which is an abundant transcript) does not mean that KDR is negative in most cells. However, the mostly negative KDR surface protein expression implies that.
3) What were the differences observed between the hESC and hiPSC line?
4) Where are scale bars e.g. in Fig. 3B or Fig. 4D right panel? What is the scale of the bars e.g. in Figs. S1A, B, C and S13C, E?
5) Please, fully specify PDGFRA in the text (line 300).
6) Catalog numbers or preferably RRIDs for all antibodies should be provided.
Reviewer 3 Report
The paper is well written, and the results are understandable. However, the concept of 3D cell culture should be introduced because the 3D culture system is the most important concept. The authors should discuss the results by comparing the related references. The paper would be accepted in Cells only when the below comments are responded.
- 58-68.
The authors should describe the concept or recent technology of the 3D cell culture system, such as the application of cell transplantation or drug research model. I suggest these reviews be added for readers’ better understanding.
Stem cells 2018;36:1329-1340 for stem cell research
Cancers 2020, 12(10), 2754 for tissue engineering technology
- Results
The time course of cell numbers should be investigated. If the authors cannot, the point should be described in the Discussion section.
Round 2
Reviewer 1 Report
In some degree, the authors addressed the issues I raised.
Author Response
Thank you for your time
Reviewer 2 Report
The manuscript improved and most of my concerns were addressed. However, since the text is still sloppy, it should be checked and proof-read. For example, the paper still contains typos and misses cross references for several Figure panels in the main text (Figs. 2B, 4C, S1 etc.).
The authors contradict themselves in lines 243-247: what are CD34+KDR+CD31+ cells according to their definition: non-hematopoietic (line 243) or hemogenic endothelial cells (lines 247, Fig. 3B)?
Author Response
Comments Reviewer 2
The manuscript improved and most of my concerns were addressed. However,
since the text is still sloppy, it should be checked and proof-read. For
example, the paper still contains typos and misses cross references for
several Figure panels in the main text (Figs. 2B, 4C, S1 etc.
Response: Agreed, we have addressed this issues, supplementary figures are now referenced in the text and we included the missing references to figures, we also change erroneous information and typos
The authors contradict themselves in lines 243-247: what are CD34+KDR+CD31+
cells according to their definition: non-hematopoietic (line 243) or
hemogenic endothelial cells (lines 247, Fig. 3B)?
Response: Agreed, the paragraph has been modified to
The cytometry analysis results described in figure 3B showed that only the suspension (3D/EB) culture format produced a population of CD34+KDR+ cells. When further analyzing this double-positive subset, we found that this cell population expresses almost in its entirety CD31 (97.5%). This suggested the emergence of a hemogenic endothelial population (CD34+KDR+CD31+). Simultaneously, we found evidence (lower panel) of a population that is positive for CD34, and positive for CD31, but negative for KDR, a possible indication of non-hematopoietic endothelial cells, showing that hematopoietic and non-hematopoietic cells can arise simultaneously during this period of differentiation.
Reviewer 3 Report
The manuscript has been improved for publication.
Author Response
Thank you for your time